# Concurrent Activation of Both Survival-Promoting and Death-Inducing Signaling by Chloroquine in Glioblastoma Stem Cells: Implications for Potential Risks and Benefits of Using Chloroquine as Radiosensitizer

**DOI:** 10.3390/cells12091290

**Published:** 2023-04-30

**Authors:** Andreas Müller, Patrick Weyerhäuser, Nancy Berte, Fitriasari Jonin, Bogdan Lyubarskyy, Bettina Sprang, Sven Rainer Kantelhardt, Gabriela Salinas, Lennart Opitz, Walter Schulz-Schaeffer, Alf Giese, Ella L. Kim

**Affiliations:** 1Experimental Neurooncology Group, Clinic for Neurosurgery, Johannes Gutenberg University Medical Centre, 55131 Mainz, Germany; mandre@students.uni-mainz.de (A.M.);; 2Institute of Toxicology, Johannes Gutenberg University Medical Centre, 55131 Mainz, Germany; patrickwey@web.de; 3NGS Integrative Genomics Core Unit (NIG), Institute for Human Genetics, University Medical Centre, 37075 Göttingen, Germany; gsalina@gwdg.de; 4Functional Genomics Center Zurich, ETH Zurich, University of Zurich, 8092 Zurich, Switzerland; lennart.opitz@fgcz.ethz.ch; 5Department of Neuropathology, Medical Faculty, Saarland University, 66123 Homburg, Germany; walter.schulz-schaeffer@uks.eu

**Keywords:** chloroquine, glioblastoma radiosensitization, glioblastoma stem cells, p53, p21-DREAM, ATM, AKT, HIPK2

## Abstract

Lysosomotropic agent chloroquine was shown to sensitize non-stem glioblastoma cells to radiation in vitro with p53-dependent apoptosis implicated as one of the underlying mechanisms. The in vivo outcomes of chloroquine or its effects on glioblastoma stem cells have not been previously addressed. This study undertakes a combinatorial approach encompassing in vitro, in vivo and in silico investigations to address the relationship between chloroquine-mediated radiosensitization and p53 status in glioblastoma stem cells. Our findings reveal that chloroquine elicits antagonistic impacts on signaling pathways involved in the regulation of cell fate via both transcription-dependent and transcription-independent mechanisms. Evidence is provided that transcriptional impacts of chloroquine are primarily determined by p53 with chloroquine-mediated activation of pro-survival mevalonate and p21-DREAM pathways being the dominant response in the background of wild type p53. Non-transcriptional effects of chloroquine are conserved and converge on key cell fate regulators ATM, HIPK2 and AKT in glioblastoma stem cells irrespective of their p53 status. Our findings indicate that pro-survival responses elicited by chloroquine predominate in the context of wild type p53 and are diminished in cells with transcriptionally impaired p53. We conclude that p53 is an important determinant of the balance between pro-survival and pro-death impacts of chloroquine and propose that p53 functional status should be taken into consideration when evaluating the efficacy of glioblastoma radiosensitization by chloroquine.

## 1. Introduction

Glioblastoma (GB) is the most malignant form of brain tumors in adults. With ~15 months survival, GB is associated with one of the poorest clinical outcomes among all human cancers [1,2,3,4]. The international standard of care for GB encompasses surgical debulking, followed by hypofractionated radiation (frRT) combined with the alkylating agent temozolomide (TMZ) [5,6]. A high degree of intrinsic and acquired radioresistance is the major challenge to anti-GB therapy with the need for effective radiosensitizing treatments remaining unmet. GBs are genetically complex tumors characterized by multiple alterations in key pathways that control cellular fate via diverse but functionally overlapping molecular mechanisms [7,8]. The biological paradigm of GB is based on a premise that malignant progression of GB is driven by so-called glioblastoma stem-like cells (GSCs), implicated as the major cellular determinants of GB radioresistance and tumor re-growth after (or under) cytotoxic treatments [9]. Owing to stemness properties and high degree of intrinsic plasticity, manifested in the ability to switch reversibly between distinct cellular states, GSCs are capable of adapting and surviving the impacts of cytotoxic treatments that are otherwise effective against glioma cells lacking stemness properties. Augmented DNA damage response (DDR) and highly efficient DNA repair is of particular importance for the ability of GSCs to withstand clinically relevant exposure to ionizing radiation (IR) [10]. There is a growing consensus that interference with GSC-associated radioresistance is a prerequisite for improving clinical outcomes in patients with GB [9,11]. In this regard, the pleiotropic agent chloroquine (ClQ) has attracted considerable attention for its potential radiosensitizing effects [12,13]. There is some clinical evidence that addition of ClQ to standard therapy can improve clinical outcomes in patients with newly diagnosed and recurrent GB [14,15,16,17]. While clinical studies are underway to evaluate the potential merits of ClQ as a radio- or chemosensitizing agent for GB [18,19,20,21,22,23,24,25,26], there is still a lack of certainty about the molecular and cellular mechanisms underlying anti-tumor actions of ClQ. Although there is considerable experimental evidence indicating that ClQ inhibits glioma cell growth in vitro, the molecular factors that determine glioma cell responsiveness to ClQ remain elusive. Up until recently, autophagic inhibition was widely considered to be the major mechanism underlying cancer cell death from ClQ. However, this view has been recently challenged by the results from independent studies addressing the relationship between the tumor-suppressing effects of ClQ and its ability to inhibit autophagy [27,28,29]. Large-scale investigations conducted by Pfizer, Novartis and AstraZeneca have provided compelling evidence that ClQ-induced cytotoxicity is unrelated to the autophagic inhibition by ClQ [28,29]. These unexpected findings reinforce efforts towards elucidating the mechanisms of ClQ-mediated radiosensitization and its determining molecular factors in particular. Activation of the p53 pathway has been implicated as one of the molecular mechanisms of ClQ-mediated cytotoxicity in different types of cancer cells, including non-stem glioma cells [30,31,32,33,34]. The impacts of ClQ on GSCs and the role of the p53 pathway in ClQ-mediated cytotoxicity are less clear. Considering that p53 is mutationally inactivated in more than 50% of GBs [35] and that some TP53 mutations result in gained activities distinct from those of the wtp53 protein [36,37,38,39], an important question is whether the mutational status of p53 has a role in determining cellular outcomes elicited by ClQ. Another source of uncertainty in translating experimental data into clinical practice is the diversity of cellular models and experimental conditions used in different studies. Up till now, ClQ’s potential to suppress glioma cells growth has been evaluated exclusively in vitro using variable cell culture conditions, mostly those that promote loss of stemness. Furthermore, the radiosensitizing potential of ClQ has been evaluated in the context of a single exposure to relatively high doses of ionizing radiation (IR) using experimental regimens that differ from multifractionated low-dose radiotherapy for GB [40]. Considering that radiation dose is a critical parameter of tumor radioresponsiveness, evaluation of the radiosensitizing potential of ClQ in vivo using clinically relevant doses of radiation is a matter of clinical importance. 

In this study, in vitro and in vivo responses mediated by ClQ alone or in combination with clinically relevant doses of IR were evaluated in patient-derived GSCs differing for the status of p53. 

## 2. Materials and Methods

### 2.1. Cells and Cell-Based Assays

Human GSC lines used in this study derive from newly diagnosed GB tissues and have been extensively characterized in previous studies [41,42,43,44,45]. The GSC line #993 has an intact TP53 gene whereas #1095 GSCs carry a nonsense mutation, generating a premature stop codon at position 146 (Appendix A). G112SP GSCs have been isolated from a glioblastoma cell line G112 and carry a hot-spot mutation R273H [46]. All GSCs used in the study possess the capacity to self-renew in vitro (Appendix A), capable of initiating and sustaining tumor growth in vivo and recapitulating a highly invasive phenotype of GB (Appendix A). In vitro cultivation of GSCs was performed using NeuroBasal medium, supplemented with the B27 component (Invitrogen Life technologies, Carlsbad, CA, USA), basic fibroblast growth factor (bFGF) and epidermal growth factor (EGF) (10 and 20 ng/mL, respectively, Biochrom GmbH, Merck KGaA, Darmstadt, Germany). For immunofluorescence staining, GSC spheres were dissociated using Trypsin (Gibco, Thermo Fischer Scientific, Darmstadt, Germany, Cat. #25300-054) diluted at a 1:1 ratio in NeuroBasal medium, plated on ornithin-coated glass coverslips at 30.000 cells/coverslip and allowed to adhere for at least 24 h. Cells were fixed with 4% paraformaldehyde, washed 3 times with PBS and incubated in blocking solution (0.1% Triton X-100, 1% bovine serum albumin) and stained overnight at +4 °C with primary antibodies diluted in blocking solution. Primary antibodies used in the study include α-Ki67 (Abcam, Cambridge UK, ab16667), α-nestin (Abcam, Cambridge UK, ab22035), and α-GFAP (Dako, Hamburg, Germany, Z0334). Secondary antibodies were goat α-mouse Alexa Fluor 488 (Invitrogen, Carlsbad, CA, USA, A-11001, 1:10,000) or goat α-rabbit Alexa Fluor 555 (Invitrogen, Carlsbad, CA, USA, A-21429). 

### 2.2. Cell Treatments and Cell-Based Assays

Single cell suspensions of GSCs were prepared one day before treatment and treated with freshly prepared ClQ (Sigma-Aldrich, St. Louis, MO, USA, C6628-25G) at 30 µM/L for desired time points. For in vitro irradiation, cells were subjected to 2.5 Gy of X-rays using a Gulmay RS225 GS014 X-ray machine (Gulmay Medical Ltd., Camberley, UK) at a dose rate of 1 Gy/min. For flow cytometry, 2 × 10^5^ cells either untreated or untreated were harvested at indicated time points after treatment, washed in PBS and pelleted at 1500 rpm and +4 °C for 8 min. Washed cells were re-suspended in PBS, fixed with ice-cold 70% ethanol on the vortex and incubated for 15 min on ice. For staining, cells were rehydrated in PBS, stained using the FxCycle™ PI/RNAse Staining Solution Kit (Invitrogen) and processed for flow cytometric analysis of DNA content using the FACSCanto™Iia (BD, Franklin Lakes, NJ, USA) instrument. Statistical analyses were performed using the GraphPad Prism Version 7 (GraphPad Software Inc., San Diego, CA, USA). Self-renewal was assessed by using the extreme limited dilution assay (ELDA) [47]. 

### 2.3. Animal Experiments

Animal experiments were conducted in accordance with the guidelines of the European Convention for the Protection of Vertebrates Used for Scientific Purposes under the permission from the State Office of Lower Saxony (permission #33.942502-04/012/07) and State Office of chemical investigations of Rhineland-Palatinate (permission #23 177-07/G12-1-020). The protocol for animal experiments was approved by the Central Animal Research Facility (ZTE) of the University Medical Centre of Göttingen and Translational Animal Research Center (TARC) of the Johannes Gutenberg University Medical Centre of Mainz. Intracranial implantation using immunodeficient mice (NMRI, female, 5–6 weeks old, Charles River Europe) was performed under standardized conditions as described previously [30,42,48]. In brief, single cell suspensions were PBS-washed and re-suspended in PBS at 3 × 10^4^ cells/µL. Cell viability was ascertained by using the trypan blue assay. A total of 10^5^ cells were implanted into the right brain hemisphere using a stereotactic frame (TSE Systems, Bad Homburg, Germany) at 1 mm anteroposterior axis, 3 mm lateromedial axis, and 2.5 mm vertical axis, relative to bregma. Three weeks after implantation, mice were divided into four groups (n = 12/group) and subjected to single treatments with ClQ or radiation or combined treatment with ClQ and radiation (Appendix A). ClQ treatment was performed one day before irradiation using an intraperitoneal (i.p.) injection of freshly prepared ClQ in 0.9% NaCl (14 mg/kg). Mice from the control group were injected with 0.9% NaCl without ClQ. Prior to irradiation, mice were anesthetized with an i.p. injection of avertine at 0.4 g/kg body weight, placed in a prone position and covered with 5 mm-thick lead plates containing a rectangular window to allow brain radiation while shielding the rest of body (Appendix A). Mice were subjected to selective brain radiation with 2.5 Gy using a Varian Clinac 600 C accelerator (dose rate 1 Gy/min). Radiation treatment was performed for 6 consecutive days (2.5 Gy per day). Control (“sham treatment”) or “ClQ only” groups were subjected to the same anesthesia protocol but did not receive irradiation. Mice termination was performed at the onset of tumor-associated neurological symptoms such as seizure, loss of balance, disorientation, and complete or partial paralysis. Mouse brains were isolated, fixed in 4% paraformaldehyde, embedded into paraffin blocks and sectioned for histological analyses. Brain tumors were confirmed by histological examinations of 3 µm sections using hematoxylin-eosin staining and analyzed by immunohistochemical of immunofluorescence staining using antibodies against human nestin (Invitrogen, Thermo Fischer Scientific, Darmstadt, Germany, PA5-82905), glial fibrillary acidic protein (GFAP) (DAKO, Z0334), or KI67 (Abcam; ab16667). For mouse survival studies, statistical analysis was performed using the log-rank (Mantel–Cox) test or Gehan–Breslow–Wilcoxon test. A *p* value < 0.05 was considered statistically significant.

### 2.4. Protein Analyses

Cells were lysed in SDS lysis buffer (1% SDS, 1 mM Tris, 1mM EDTA, and pH 8.0) supplemented with a protease inhibitors cocktail (cOmplete^TM^, Sigma-Aldrich) for 10 min at 95 °C and subjected to ultrasound sonification using an Ultrasonicator (Bandelin Sonopuls). After sonification, cell lysates were cleared by centrifugation at 14,000 rpm for 15 min at +4 °C. Protein concentration was determined by using the NanoDrop spectrophotometer (NanoDrop Technologies Inc, Wilmington, DE, USA). Electrophoretic protein separation was achieved by using the precast gels mini-PROTEAN^®^ TGX™ (Bio-Rad, Feldkirchen, Germany) followed by proteins transfer on a PVDF membrane (Thermo Fischer Scientific GmbH, Darmstadt, Germany). Antibodies against p53, p21, p53Ser46P, ATM, ATMSer1981P, AKT, AKTSer473P, HIPK2, LC3B-I, LC3B-II or p62 were from Cell Signaling Technology Inc., Danvers, MA, USA. Other antibodies used were a-p53Ser15P (Proteintech Germany GmbH, Planegg-Martinsried, Germany), a-HIPK2Tyr361P (MyBioSource Inc, San Diego, CA, USA), a-actin (Santa Cruz Biotechnology, Inc., Dallas, TX, USA) or a-HSP70 (Enzo Life Sciences Inc., Farmingdale, NY, USA). On-array protein analyses were performed using the human apoptosis antibody array AAH-APO-G1-8 (RayBiotech Inc., Peachtree Corners, GA, USA) according to the supplier’s recommendations. In brief, cell lysates were prepared from 2 × 10^6^ cells and incubated with the array slide overnight at +4 °C on the rotating platform to allow proteins to bind to the array. After the binding step, arrays were washed and subjected to another round of incubation with a cocktail of biotinylated antibodies against apoptosis-related proteins, washed, and incubated with fluorescence dye. Fluorescence signals were detected by using X-ray films (Fujifilm Holdings Corporation, Minato, Japan). Signal intensity was quantified by densitometry using ImageJ (https://imagej.nih.gov, accessed on 1 February 2023). 

### 2.5. Gene Expression and Bioinformatics

Gene expression was analyzed in three glioblastoma stem cell lines (#993, #1095 and G112) either untreated or treated with ClQ. In addition to ClQ-associated gene expression, transcriptomic changes associated with exposure to IR were also analyzed in line #993. For each line and each condition (treated or untreated), 3 biological replicates were analyzed. RNA isolation, processing, and array-based profiling using GeneChip^®^ Human Gene 1.0 ST Array (Affymetrix) were performed as described previously [43]. In brief, RNA was isolated using the Trizol (Invitrogen, Waltham, MA, USA) method according to the manufacturer’s instructions, treated with DNAse I (Sigma-Aldrich, St. Louis, MO, USA) and checked for quality using the Agilent 2100 Bioanalyzer (Agilent Technologies, Santa Clara, CA, USA). cDNA was synthesized using the WT Target Labeling and Control Reagents (Affymetrix, Santa Clara, CA, USA) followed by the cleanup using the GeneChip^®^ Sample Cleanup module (Affymetrix). In vitro transcription was conducted using the WT Target Labeling Kit (Affymetrix). Data analysis was performed by using the affy [49] and Limma package [50] of Bioconductor [51]. The data analysis consisted of between-array normalization, probe summary, global clustering and PCA-analysis, fitting the data to a linear model and detection of differential gene expression. To ensure that the intensities had similar distributions across arrays, quantile-normalization was applied to the log2-transformed intensity values. As for the summary of probes, a median polish procedure was chosen. Significant changes in the expression of genes between the groups was analyzed by empirical Bayes statistics by moderating the standard errors of the estimated values [52]. The *p*-values obtained from the moderated t-statistic were corrected for multiple testing with the Benjamini–Hochberg method [53]. These *p*-value adjustments guarantee a smaller number of false positive findings by controlling the false discovery rate (fdr). For each gene, the null hypothesis suggesting there is no differential expression between degradation levels was rejected when its fdr was lower than 0.05. Samples were assessed in a blinded manner. Gene expression data and results of bioinformatic analyses are deposited in the Gene Expression Omnibus database with the accession number GSE225191. Textual and graphical outputs of the results of cross-comparison between the genes expressed differentially in different experimental groups were performed by using the Whitehead BaRC (http://barc.wi.mit.edu/tools/, accessed on 1 February 2023) and Venn Diagram (https://bioinformatics.psb.ugent.be/webtools/Venn/, assessed on 1 February 2023) web tools.

### 2.6. Statistical Analysis

In vitro experiments were performed at least three times. Data were evaluated using Student’s *t* test and presented as mean ± SD. *p* values ≤ 0.05, ≤ 0.01, ≤ 0.001 or ≤ 0.0001, respectively, were considered statistically significant, very significant, highly significant or most significant. For mouse survival studies using the Kaplan–Meier method statistical analysis was performed using the log-rank test or Gehan–Breslow–Wilcoxon test. A *p* value < 0.05 was considered statistically significant. All statistical analyses were performed using GraphPad Prism (version 6.01).

## 3. Results

### 3.1. Impact of ClQ Alone or in Combination with IR on GSCs Proliferation and Viability In Vitro

ClQ suppresses in vitro proliferation and viability of non-stem glioma cells in the range of 20–50 µM [30,54,55,56]. To enable a direct comparison with previously published data, ClQ at 30 µM was used for in vitro treatments of GSCs throughout the study. Similarly with the effects of ClQ reported for non-stem glioma cells, ClQ effectively inhibits in vitro proliferation in all GSCs tested (Figure 1 and Appendix A). 

The results showed that the degree of ClQ-mediated proliferative inhibition in vitro is comparable or even greater than that induced by clinically relevant doses of ionizing radiation (2.5 Gy). Cell death assessments, using the sub-G1 assay, revealed that ClQ treatment leads to the increase in fractional DNA content in all GSCs tested, albeit to varying degrees, with wtp53 GSCs being the least susceptible to ClQ-induced cell death compared to R273H-p53 and p53 null GSCs (Figure 2 and Appendix A). Vice versa, wtp53 GSCs showed greater radiosensitivity in vitro than R273H-p53 or p53 null GSCs as evidenced by a moderate but significant increase in sub-G1 cells after IR in wtp53 GSCs but not in R273H-p53 or p53 null GSCs. Notably, the combined treatment with ClQ and radiation in vitro failed to significantly augment cell death rates, which were even reduced in wtp53 and p53 null GSCs compared to ClQ treatment alone (Figure 2). R273H-p53 GSCs did show a trend toward increased cell death after combined treatment with ClQ and IR but the difference between ClQ+IR and ClQ alone treatments was not statistically significant (Figure 1 and Appendix A). Together, these data demonstrate that ClQ is effective in both inhibiting GSC proliferation and inducing GSC death in vitro and indicate that GSCs differing in the status of p53 vary in their susceptibility to ClQ either in a solo setting or in combination with IR. 

### 3.2. In Vivo Effects of ClQ Alone or in Combination with IR on the Tumor-Propagating Capacity of GSCs

We next sought to determine if the proliferation-inhibiting and death-promoting impacts of ClQ, effective against in vitro propagating GSCs, can be realized in vivo and suppress GSC-driven tumor growth. To that end, ClQ impacts, alone or in combination with clinically relevant doses of hypofractionated IR (frIR, six daily fractions of 2.5 Gy), were evaluated in GSC xenografts as depicted in Appendix A. In vivo outcomes of frIR treatment showed good concordance with the patterns of GSC radiosensitivity in vitro. Consistent with in vitro radioresistance of p53-R273H and p53 null GSCs (Figure 1), their derived tumors also manifest a radioresistant phenotype (Figure 3a), whereas wtp53 GSCs, showing a greater higher degree of radiation-induced death in vitro (Figure 2), recapitulate a radiosensitive phenotype in vivo (Figure 3a). Unlike frIR, ClQ treatment in vivo poorly reproduces ClQ’s efficacy in inhibiting GSC proliferation and viability in vitro as evident from the lack of significant effect of ClQ treatment on xenografted GSCs irrespective of their p53 status (Figure 3b). Moreover, tumors grown from p53-R273H GSCs showed a tendency to grow even faster after the treatment with ClQ (Figure 3b). However, the combined treatment with ClQ and frIR proved effective in retarding p53-R273H tumors as evidenced by a significant (*p* = 0.001) prolongation of survival compared to the sham-treated control group (Figure 3c) or mice treated singly with ClQ or frIR (Figure 3a,b). 

In contrast to p53-R273H xenografts, wtp53 or p53 null xenografts showed no significant prolongation of survival after combination treatment with ClQ and frIR compared to single treatments with ClQ or frIR. Together, our in vivo data indicate that (i) ClQ is poorly effective in inhibiting the tumor-propagating capacity of GSCs in a single treatment setting; (ii) combination treatment with ClQ and frIR is effective in sensitizing GSC tumors to radiation, but in a differential manner; and (iii) wtp53 does not render GSC tumors more susceptible to ClQ-mediated radiosensitization.

### 3.3. ClQ Elicits Distinct Molecular Outcomes in GSCs Differing for the Status of p53

The p53-dependent and p53-independent mechanisms have been implicated in ClQ-induced death of cancer cells including non-stem glioma cells [30,57,58,59]. To address the relationship between ClQ and p53 activity in GSCs, we assessed the effects of ClQ on the p53 protein and p53 modulating factors in GSCs expressing wtp53 or R273H-p53 mutant. The results showed that ClQ treatment leads to the accumulation of p53 protein and its transcriptional target p21 in wtp53 GSCs but not in GSCs with transcriptionally impaired mutant R273H-p53 (Figure 4).

Surprisingly, p53 accumulation induced by ClQ is unaccompanied by p53 phosphorylation on key regulatory serines, including Ser46 (Figure 4) or phosphorylation of a p53-specific E3-ligase MDM2 on Ser395, a modification that is required for p53 stabilization after DNA damage (Figure 5). 

Lack of p53 and MDM2 phosphorylation on serines 46 and 395, respectively, in spite of p53 accumulation induced by ClQ is intriguing considering that these modifications are mediated by the ataxia-telangiectasia mutated (ATM) protein kinase, which is sensitive to ClQ [60] and phosphorylates both p53-Ser46 of p53 and MDM2-Ser395 in response to DNA damage [61,62,63]. We next sought to clarify if the canonical signaling involved in p53 accumulation induced by DNA damage also operates in ClQ-treated GSCs. While keeping in mind that ClQ treatment induces an activating ATM autophosphorylation on Ser1981 [60] the levels of ATM-Ser1981P were assessed in untreated and ClQ-treated GSCs. Consistent with previous findings that constitutive activation of ATM is a hallmark of GSCs [10], untreated GSCs show considerably higher steady-state levels of ATM-Ser1981P than non-stem glioblastoma cells U87 in which ATM-Ser1981P is barely detectable in the absence of DNA damage but increases upon exposure to radiation (Appendix A). 

Strikingly, GSCs treated with ClQ showed a dramatic change in the ATM-Ser1981P pattern characterized by the appearance of smaller ATM-Ser1981P (termed hereafter as ΔATM-Ser1981P) with the apparent molecular weight of ~250 and 100 kDa (Figure 6). Confirming their ATM origin, ΔATM-Ser1981P forms are recognized by both pan-ATM and ATM-Ser1981P-specific antibodies (Appendix A). Notably, the truncated ATM forms make a major contribution to the overall increase in Ser1981 phosphorylation levels (Figure 6) indicating that ClQ-induced fragmentation affects preferentially the active, ATM-Ser1981P form. These data thus reveal a duality of ClQ impacts on ATM: on the one hand, ClQ induces an autoactivating phosphorylation of Ser1981 confirming the previous findings of Bakkenist and Kastan [60] but on the other hand, it also promotes ATM-Ser1981P fragmentation, a previously unknown action of ClQ that occurs in all GSCs tested albeit with varying degrees and temporal dynamics (Figure 6 and Appendix A). ClQ-induced fragmentation of ATM-Ser1981P is a phenomenon specifically associated with ClQ because it does not occur in radiation-treated GSCs (Appendix A). 

It has been shown that ATM fragmentation by proteolytic cleavage is one of the mechanisms inactivating its kinase activity towards Ser46 of the p53 protein in particular [64]. In the light of this knowledge, ClQ-induced fragmentation of ATM might provide a plausible explanation for why wtp53 accumulation induced by ClQ in GSCs is unaccompanied with p53 phosphorylation on Ser46 (Figure 4). However, ATM is not the only kinase that phosphorylates p53 on Ser46, which is also targeted by the homeodomain-interacting protein kinase 2 (HIPK2). HIPK2-mediated phosphorylation of p53-Ser46 is of special interest because it plays a crucial role in transcription of p53-regulated apoptotic genes. Therefore, we have also examined the effect of ClQ on HIPK2. Western blot assessments showed that ClQ treatment leads to a marked decline in the levels of both total HIPK2 and its activated form HIPK2-Tyr361P, which occurred in all tested GSCs irrespective of their p53 status (Figure 7). 

Altogether these results reveal that in addition to its potential to induce p53 accumulation ClQ also reduces the abundance of p53-regulatíng kinases ATM and HIPK2 that occurs in GSCs either proficient or deficient for the wtp53 function. 

### 3.4. ClQ Induces Transcriptional Repression via the p53-p21-DREAM Pathway Concurrently with Transcriptional Activation of the Mevalonate Pathway

ATM and HIPK2 kinases are important modulators of p53-mediated transcription, encompassing hundreds of genes [65]. Our finding that ClQ affects the abundance of active ATM and HIPK2 (Figure 6 and Figure 7) prompted us to test the effects of ClQ on p53-dependent transcription. To that end, a microarray-based approach was employed to identify differentially expressed genes influenced by ClQ (termed as “ClQ_DEGs”). Gene expression analyses revealed that ClQ treatment elicits considerable transcriptional changes in all tested GSCs albeit in varying degrees (Figure 8). 

Transcriptional changes induced by ClQ are considerably more profound in wtp53 GSCs and p53-R273H GSCs (144 and 180 differentially expressed genes, respectively, Appendix A) than in p53 null GSCs (24 genes, Appendix A). The functional spectrum of ClQ_DEGs also differs markedly between GSCs. In wtp53 GSCs, the majority of ClQ_DEGs (up-regulated and down-regulated) is constituted by p53 regulated genes (p53RGs), indicating a robust induction of p53-dependent transcriptional response (Figure 8 and Table 1). The predominance of p53RGs is especially pronounced among down-regulated ClQ_DEGs, comprised nearly completely by genes from the p21-DREAM pathway, the major mechanism for cell cycle control via p53-dependent transcriptional repression [66,67,68]. Also, among up-regulated genes, almost half of ClQ_DEGs are p53RGs from the mevalonate (MVA) pathway associated with cancer cells survival (Table 1), whereas apoptosis-inducing p53RGs are not activated by ClQ.

Consistently with impairment of transcriptional activity in mutant p53 proteins, p53RGs constitute only a minor fraction of all ClQ_DEGs identified in GSCs that express R273H-p53 (Table 2). 

In p53 null GSCs, the impact of ClQ on transcription is the weakest in terms of both the overall number of ClQ_DEGs and their dependence on p53 activity, with none of the ClQ_DEGs identified in p53-null GSCs being a known p53RG (Table 3). 

Notably, more than half of up-regulated and nearly all down-regulated ClQ_DEGs identified in p53-null GSCs are known to be associated with GB progression, resistance to therapy, or GB stemness (Appendix A). Collectively, gene expression analyses reveal that (i) ClQ has an influence on both p53-dependent and p53-independent transcription; (ii) p53 status has a decisive impact on the functional spectrum of ClQ-modulated genes; and (iii) p53-regulated genes with pro-survival functions are preferentially impacted by ClQ in wtp53 GSCs.

### 3.5. Impacts of ClQ on the Apoptotic Signaling 

Our gene expression analysis indicates that apoptosis mediated via p53-dependent transcription is unlikely to be the mechanism of ClQ-induced cell death. To gather further insights into the impacts of ClQ on apoptotic signaling, we made use of the Apoptosis Signaling Array (AAH-APOSIG1, Appendix A). The results showed that ClQ treatment elicits changes in the apoptotic signaling in both wtp53 and R273H-p53 GSCs, albeit to varying degrees. For example, caspase-3 cleavage is quite profound in wtp53 GSCs (Figure 9a) but considerably less effective in R273H-p53 GSCs (Figure 9b). One common change induced by ClQ comparably strongly in wtp53 GSCs and R273H-p53 GSCs is a marked reduction in AKT-Ser473P, an activated form of survival-promoting kinase AKT and its downstream target BAD-Ser112P.

This effect was confirmed by western blot assessments showing that all tested GSCs, irrespective of their p53 status, undergo a drastic decline in the abundance of active AKT, AKT-Ser473P, upon the treatment with ClQ (Figure 10). 

The reduction in AKT levels caused by ClQ occurred in all GSCs tested, indicating that this activity of ClQ is independent on the status of p53 and probably is conserved across different molecular subtypes of GSCs. Considering that AKT is the key factor promoting GB radioresistance, its reduction in ClQ-treated GSCs suggests that this mechanism might be involved in ClQ-mediated radiosensitization of glioma cells. This hypothesis is consistent with previous findings showing that AKT inhibition in conjunction with ClQ treatment sensitizes non-stem glioma cells to apoptosis [69], an effect that has been explained by autophagy-related impacts of ClQ known to impair the fusion between autophagosomes and lysosomes [69]. With this knowledge in mind, we sought to determine if ClQ-induced reduction in AKT levels occurs concurrently with the autophagic inhibition in GSCs. To that end, autophagy p62 and LC3BII were assessed in untreated and ClQ-treated GSCs. The results showed that ClQ treatment leads to a robust increase in both p62 and LC3BII in all GSCs tested (Figure 11). 

These results indicate that ClQ-mediated blockage of autophagy and reduction in the abundance of active AKT occur concurrently in genetically distinct backgrounds and constitute conserved traits of ClQ actions in contrast to the variable effects of ClQ on transcription (Figure 8).

## 4. Discussion

This study reports on molecular and cellular outcomes elicited by a putative radiosensitizer ClQ in human GSCs in vitro and in vivo. Our investigations reveal a high degree of functional and mechanistic versatility of ClQ in concurrent activation of pro-survival and pro-death signaling via transcriptional alterations and direct impacts on the abundance of proteins involved in cell fate determination. We provide evidence indicating that p53 status has a decisive impact on transcriptional changes induced by ClQ. In GSCs with wtp53, the transcriptional response induced by ClQ is primarily determined by p53-dependent transcriptional repression via the p53-p21-DREAM pathway [66,68] and p53-dependent transcriptional activation of glioma-promoting genes from the mevalonate pathway [70,71]. Our data reveal that ClQ-activated p53-dependent transcription differs in several aspects from the canonical mechanism of p53 activation by DNA damage. One is that p53 accumulation induced by ClQ does not involve post-translational modifications essential for p53 stabilization after DNA damage, such as phosphorylation of a p53-specific E3-ligase MDM2 on Ser395 [72,73]. Nor does it involve p53 phosphorylation on Ser46, a modification that promotes p53-dependent activation of apoptotic genes. We provide a mechanistic explanation for the lack of these regulatory modifications by uncovering a previously unknown impact of ClQ on the abundance of p53 regulatory kinases ATM and HIPK2, which are required for the execution of p53-dependent apoptosis [74,75,76]. The duality of ClQ manifests in its ability to induce p53 accumulation and to affect the abundance of ATM and HIPK2 at the same time. As these kinases are essential promoters of p53-dependent apoptosis, their decreased levels in ClQ-treated GSCs predict a selective weakening of pro-death but not pro-survival responses mediated by p53 via transcriptional regulation. Apart from its impacts on ATM and HIPK2, ClQ also affects the abundance of anti-apoptotic kinase AKT in company with inhibiting autophagy, a condition that induces death in non-stem glioma cells [69]. While our data support the hypothesis that concurrent inhibition of AKT and blockage of lysosomal degradation may promote glioma cells death [69], they also indicate that this mechanism is insufficient to suppress GSC-driven tumor growth in the absence of radiation. However, in the context of radiation treatment, the potential of ClQ to suppress tumor growth can become realized if its pro-survival activities mediated via wtp53 are “neutralized” by inactivating mutations in the TP53 gene. A similar conclusion has been reached in the study by Palanichamy et al., showing that GB radiosensitization via AKT silencing is only effective in the context of mutant p53 [77]. In the light of key AKT roles in GB radioresistance [77,78,79,80,81,82], our finding that ClQ affects the abundance of AKT supports the potential merit of ClQ as a radiosensitizing agent for GB. However, our data also indicate that ClQ-mediated radiosensitization is not a general phenomenon but a potential that can be realized depending on the p53 background, with mutated p53 emerging as a positive predictive factor for radiosensitization by ClQ. That the mere lack of p53 protein is insufficient to render GSC-driven tumors more susceptible to ClQ-mediated radiosensitization supports the view that specific activities of p53 mutant proteins should be taken into consideration when assessing the actionability of anti-cancer treatments [83]. From the mechanistic viewpoint, preferential sensitivity of GSC-driven tumors with mutated TP53 is consistent with the concept of synthetic lethality [84], whereby the combination of two inactivating events (p53 mutation and AKT inhibition, in the context of our study) is a prerequisite of effective outcome (ClQ-mediated tumor radiosensitization). The above interpretation relies on a premise that some common functions exerted by p53 and AKT should be impaired simultaneously to achieve the synthetic lethal effect. In this regard, DNA damage response (DDR) may be one of the intersecting points relevant for ClQ-mediated radiosensitization of GSC-driven tumors. Considering that DDR is the major mechanism by which both p53 and AKT protect cells from the killing effects of radiation [82,85,86,87,88], it is plausible to hypothesize that ClQ-mediated reduction in AKT in the background of DDR-impaired mutant p53 might create a condition for synthetic lethality from radiation and ClQ. As mutant p53 proteins are not simply inactive proteins but possess residual as well as diverse activities associated with “mutant p53 gain-of-function” [36,37,38], it will be important to clarify if a predisposition to ClQ-mediated radiosensitization is associated with all or only certain forms of mutant p53. 

Basing on our findings, we propose a model in which the balance between inherently antagonistic impacts of ClQ on survival-promoting or death-inducing pathways is modulated by p53 (Figure 12). 

According to this model, functionality of transcriptional p53-p21-DREAM and p53-MVA axes has a decisive impact on the prevalence of pro-survival or pro-death outcomes that can be elicited by ClQ via p53-dependent and p53-independent mechanisms. In the wtp53 background, ClQ-mediated activation of survival-promoting pathways p21-DREAM and MVA serves as a counterbalance for pro-death signals elicited by ClQ via its p53-independent impacts on the abundance of cell fate regulators including ATM, AKT and HIPK2. In the background of mutant p53 with impaired potential to activate p21-DREAM and MVA pathways, pro-survival impacts of ClQ would prevail, a condition that gains special relevance in the context of radiation-induced DNA damage. As radiation itself is a potent activator of DDR, it could be envisaged that the combined action of ClQ and radiation treatment will augment the overall DDR capacity in the context of wtp53 but not in the context of transcriptionally impaired mutated p53. Indeed, transcriptomic changes induced by ClQ or radiation in wtp53 expressing GSCs show a striking overlap in differentially expressed genes, of which most are from the p21-DREAM pathway (Appendix A). The proposed model postulates that anti-tumor potential of ClQ has clinical relevance primarily in the context of radiation treatment and that impaired DDR is an important condition for ClQ-mediated tumor radiosensitization.

## 5. Conclusions

ClQ can activate both survival-promoting and survival-inhibiting responses in GSCs via transcription-dependent and transcription-independent mechanisms. Anti-tumor potential of ClQ has clinical relevance, primarily in the context of radiation. In the absence of radiation, the potential effectiveness of ClQ as monotherapy for GB is questionable or may even lead to unwanted outcomes. The ultimate outcome of ClQ impacts in radiated GSCs is determined by a balance between pro-death and pro-survival signals, elicited at the level of transcriptional control as well as via a direct impact on abundance of cell fate regulators including p53, HIPK2, AKT, and their common modulator ATM. Our data call for caution in overinterpretation of in vitro effects of ClQ and emphasize the importance of in vivo testing with consideration of the impacts of the tumor microenvironment. Functionality of p53-p21-DREAM and p53-MVA axes has a decisive impact on the ultimate outcome exerted by ClQ with mutated p53 being a positive predictive factor ClQ-mediated radiosensitization.

## Figures and Tables

**Figure 1 cells-12-01290-f001:**
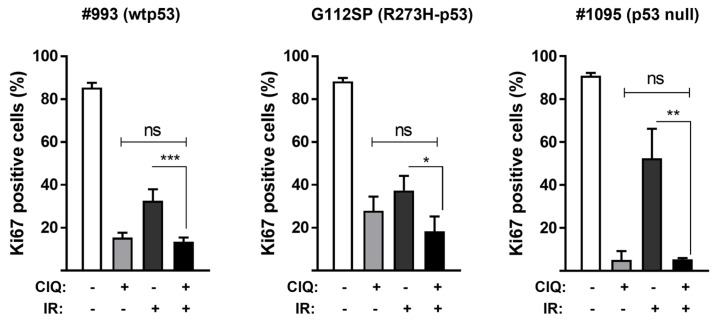
Effects of ClQ on GSCs proliferation in vitro. GSCs were treated with ClQ (30 µM), irradiation (IR, 2.5 Gy) or combination of ClQ+IR for 72 h and analyzed by immunofluorescence staining for Ki-67. Summary of the data from three independent experiments. Statistical significance was determined using Student’s *t*-test. (*), *p* ≤ 0.05; (**); *p* ≤ 0.01; (***), *p* ≤ 0.001. “ns”, not significant.

**Figure 2 cells-12-01290-f002:**
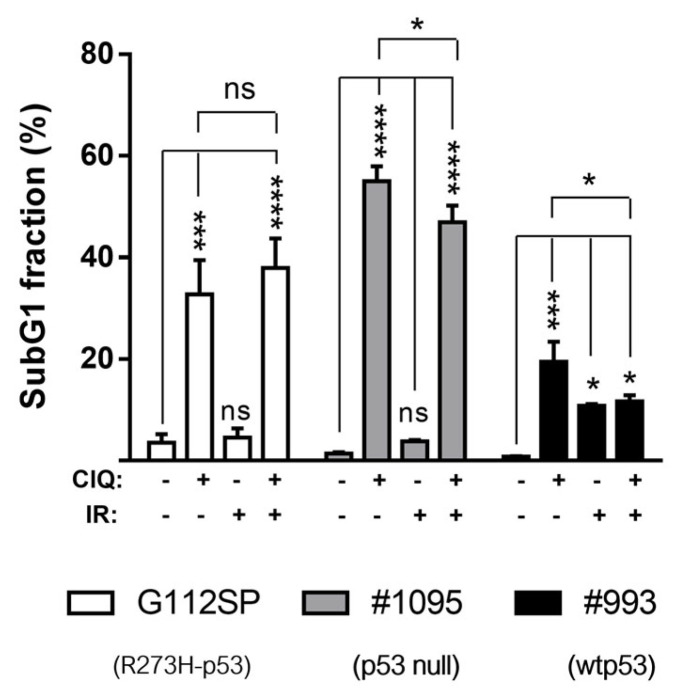
Effects of ClQ on GSCs viability in vitro. GSCs were treated with ClQ (30 µM), irradiation (IR, 2.5 Gy) or combination of ClQ+IR for 72 h and assessed for the sub-G1 content by flow cytometry. Summary of the data obtained from three independent experiments. Statistical significance was determined by an unpaired *t*-test with Welch’s correction. (*), *p* ≤ 0.05; (***), *p* ≤ 0.001; (****), *p* ≤ 0.0001, “ns”, not significant.

**Figure 3 cells-12-01290-f003:**
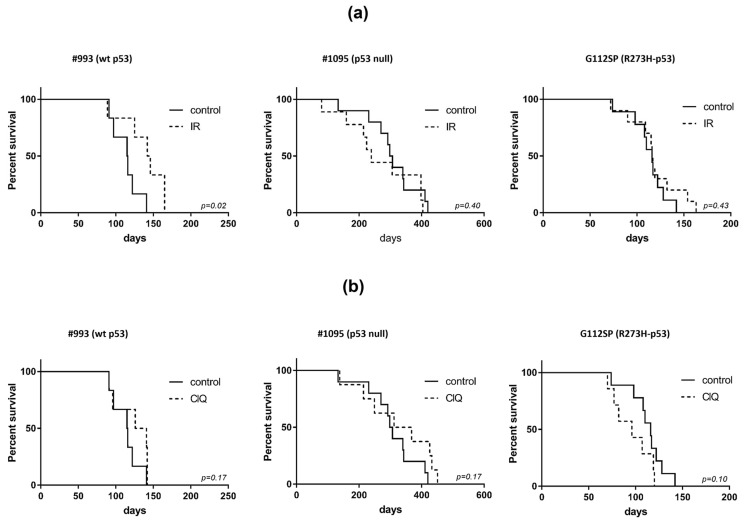
Effects of ClQ in vivo. Survival analyses of GSC xenografted mice treated with ClQ (**a**), radiation (**b**) or combination of ClQ and IR (**c**). Solid lines correspond to sham-treated control groups. Kaplan–Meier curves of mice survival were determined using the log-rank test.

**Figure 4 cells-12-01290-f004:**
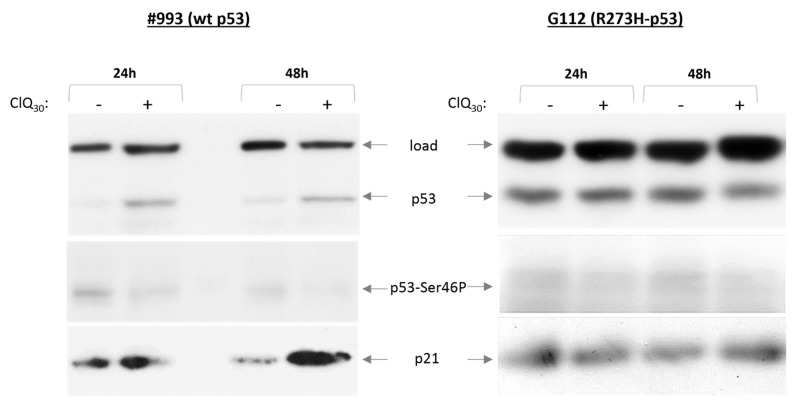
Effects of ClQ on p53, p53-Ser46P and p21 proteins. Top, representative blots for wtp53 or R273H-p53 expressing GSCs treated with ClQ for 24 h and 48 h. Protein loading was ascertained by probing for the mitochondrial resident mtHSP70. Graph shows quantitative evaluations of p53 and p21 levels by densitometry, in untreated or ClQ-treated GSCs. For total protein normalization, mitochondrial HSP70 or b-actin were used as internal loading controls. Data from three independent experiments were analyzed for each line.

**Figure 5 cells-12-01290-f005:**
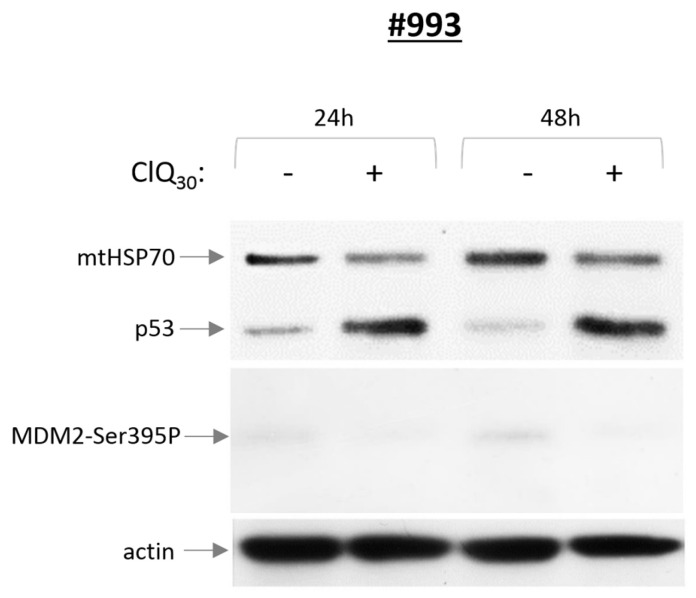
Assessment of p53 and MDM2-Ser395P proteins in wtp53 expressing GSCs. Representative blot for wtp53 GSCs treated with ClQ for 24 h and 48 h. Experiments were performed at least three times.

**Figure 6 cells-12-01290-f006:**
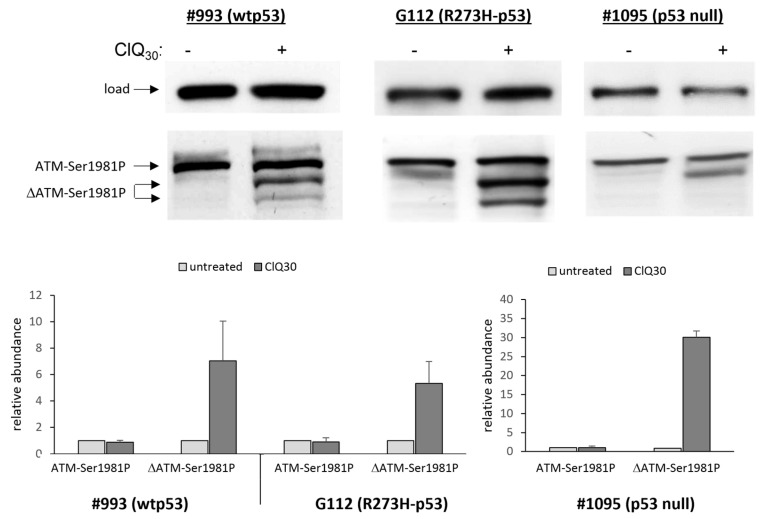
Dual effect of ClQ on ATM phosphorylation at Ser1981 and structural integrity of the ATM-Ser1981P protein. Top panel shows representative blots for ATM-Ser1981P in wtp53 (#993), R273H-p53 (G112) or p53-null GSCs after 72 h of treatment with ClQ. Graph shows the results of quantitative evaluations of the full-length and truncated ATM-Ser1891P levels by densitometry (n = 3 for each line). For total protein normalization, mitochondrial HSP70 or b-actin were used as internal loading controls.

**Figure 7 cells-12-01290-f007:**
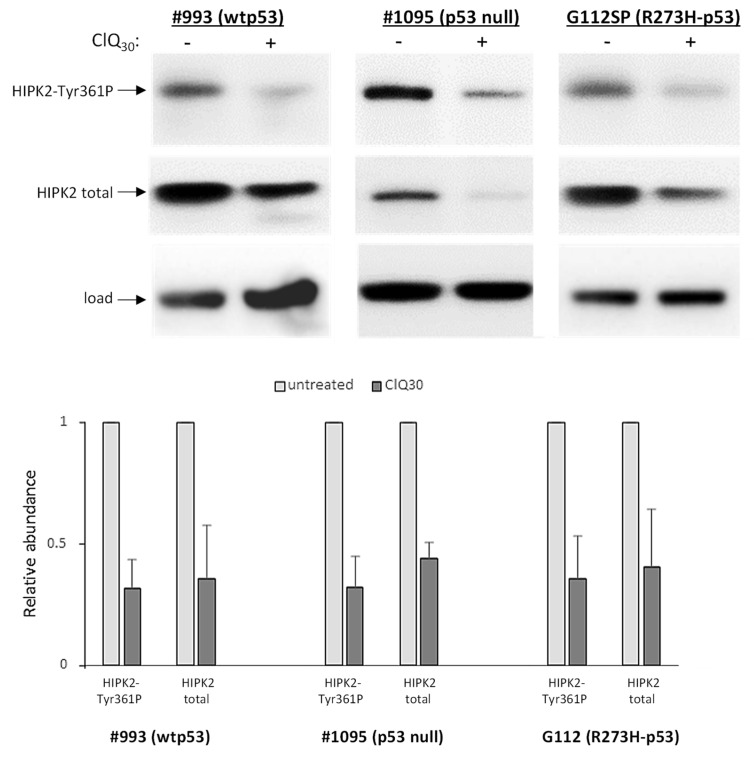
Assessments of HIPK2 proteins in GSCs differing for the p53 status. Western blot data for total and Tyr361P phosphorylated HIPK2 in GSCs expressing wtp53 (#993), R273H-p53 (G112) or p53-null GSCs after 72 h of treatment with ClQ. Top panel shows representative blots for total HIPK2 and HIPK2-Tyr361P isoform in wtp53 (#993), R273H-p53 (G112) or p53-null GSCs after 72 h of treatment with ClQ. Graph shows the results of quantitative evaluations by densitometry (n = 3 for each line). For total protein normalization, mitochondrial HSP70 or b-actin were used as internal loading controls.

**Figure 8 cells-12-01290-f008:**
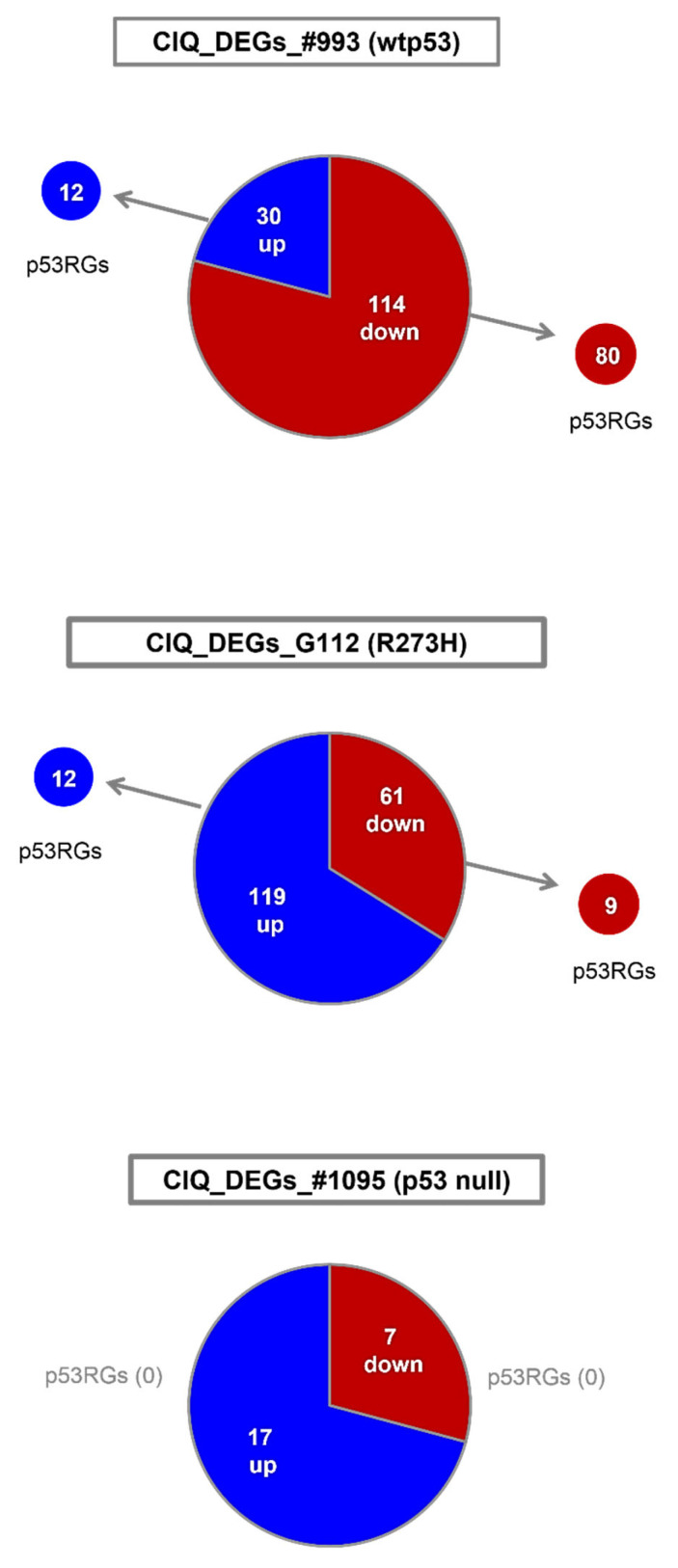
Schematic presentation of ClQ_DEGs identified in GSCs differing for the p53 status. “p53RGs”, p53-regulated genes. “up”, upregulated ClQ_DEGs. “down”, down-regulated ClQ_DEGs. Encircled numbers correspond to known p53RGs.

**Figure 9 cells-12-01290-f009:**
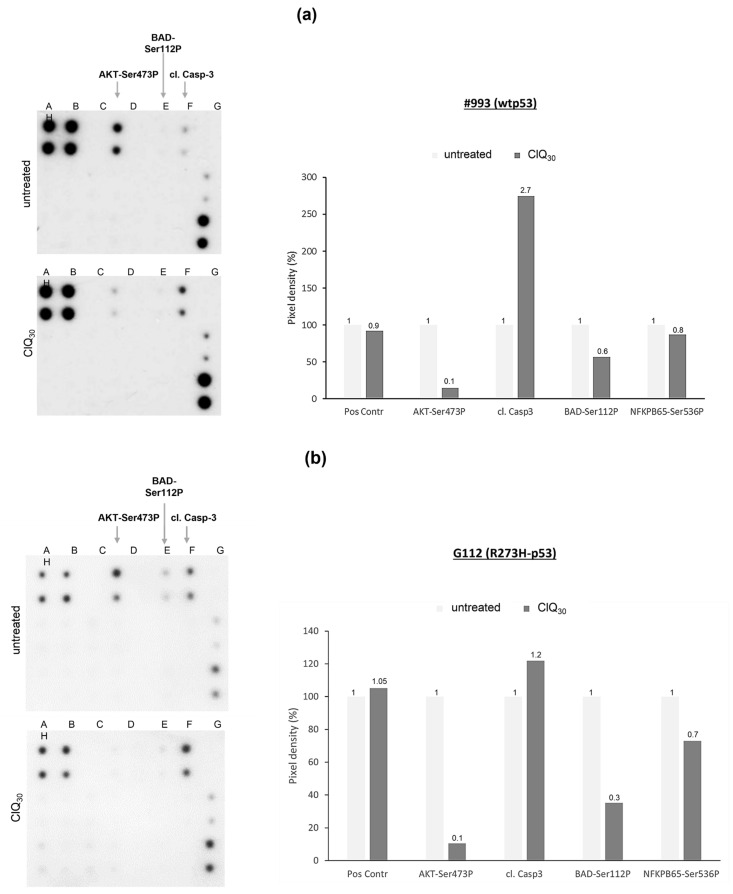
Effects of ClQ on apoptosis signaling pathways. Readouts from the APOSIG arrays incubated with cell lysates of (**a**) wtp53 or (**b**) R273H GSCs either untreated or treated with ClQ for 72 h and graphical presentation of the quantified readouts.

**Figure 10 cells-12-01290-f010:**
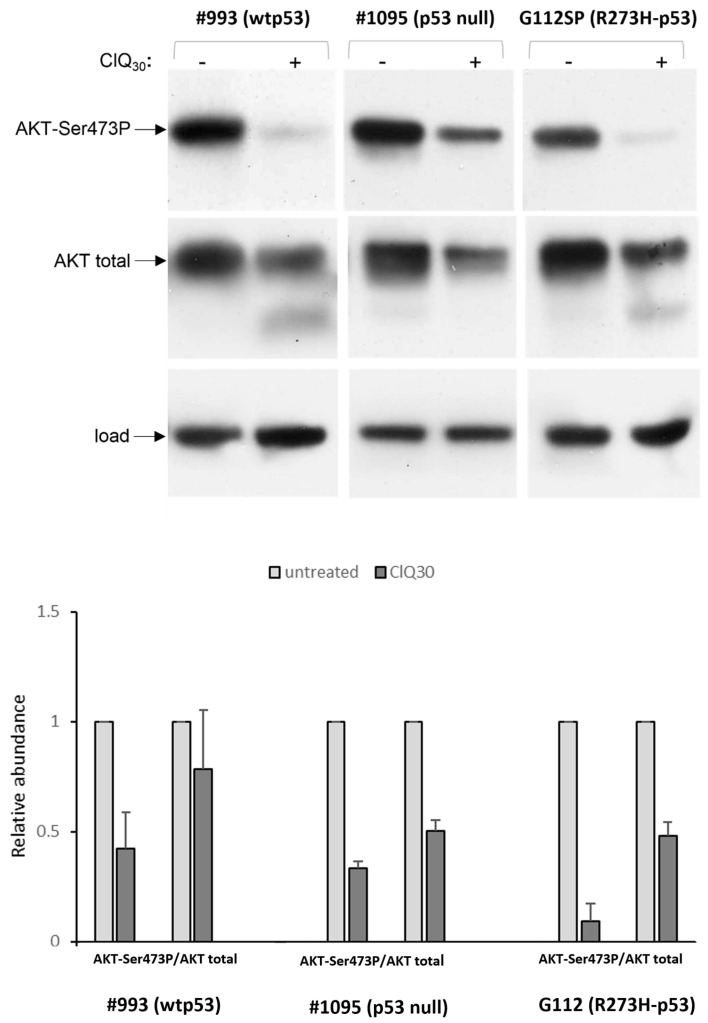
Effect of ClQ on the abundance of AKT kinase. Top panel shows representative blots for total HIPK2 and HIPK2-Tyr361P isoform in wtp53 (#993), R273H-p53 (G112) or p53-null GSCs after 72 h of treatment with ClQ. Graph shows the results of quantitative evaluations of datasets from independent experiments (n = 3 for each line) by densitometry. For total protein normalization, mitochondrial HSP70 or b-actin were used as internal loading controls.

**Figure 11 cells-12-01290-f011:**
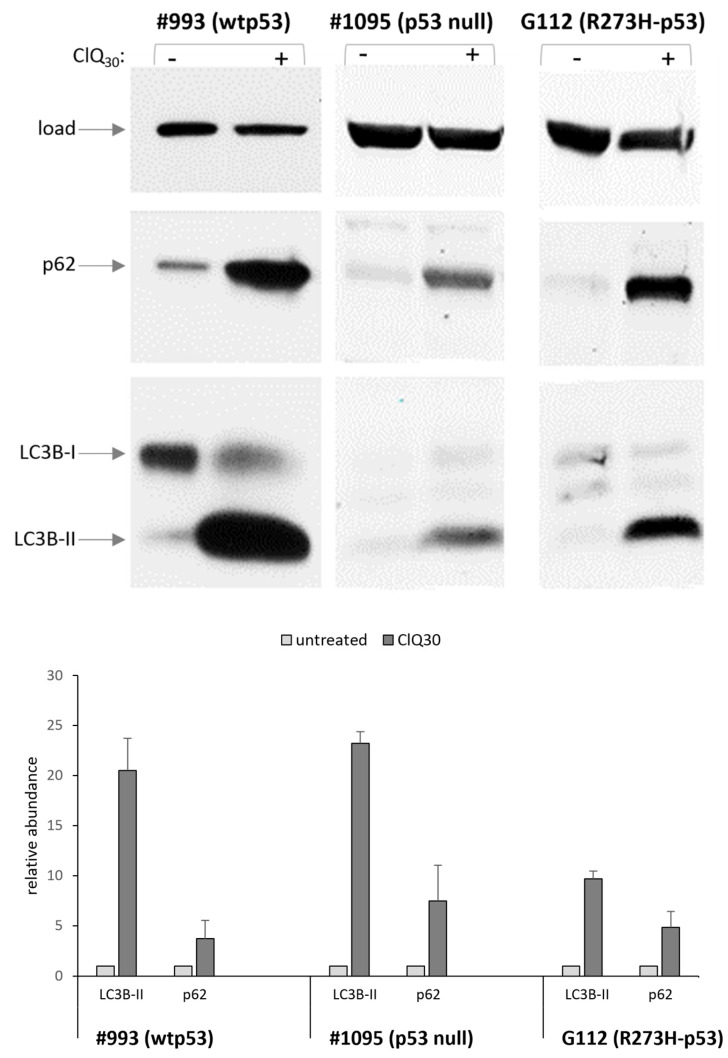
Effects of ClQ on the autophagic activity in GSCs differing for p53 status. Western blot assessments of late autophagy markers p63 and LC3B-II in untreated or ClQ-treated (72 h) GSCs. Protein loading was ascertained by probing for β-actin.

**Figure 12 cells-12-01290-f012:**
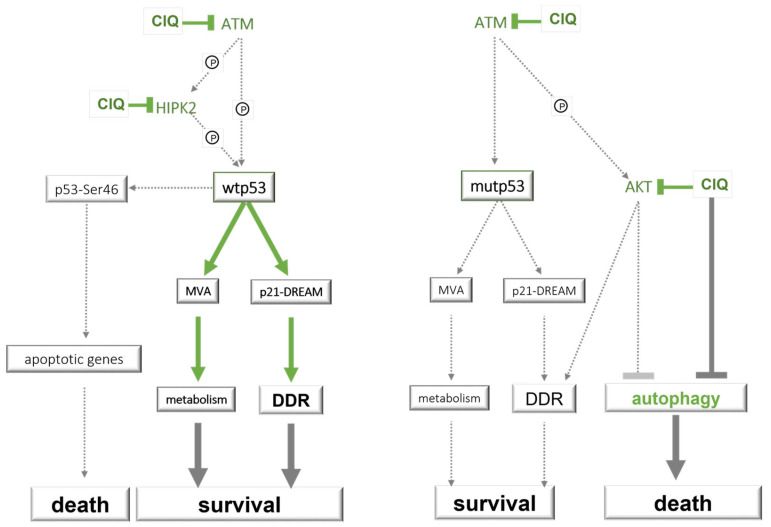
Schematic summary of main results integrated into the known networks of survival or death pathways. Green lines indicate molecular impacts of ClQ identified in this study. Solid and dashed indicate, respectively, sustained or diminished signaling in the context of wtp53 or transcriptionally impaired p53.

**Table 1 cells-12-01290-t001:** Contribution of p53-regulated genes to the transcriptomic response induced by ClQ in GSCs expressing wild type p53. p53RGs, p53-regulated genes; ClQ_DEGs, chloroquine-induced differentially expressed genes; MVA, mevalonate pathway; GO, Gene Ontology. (*), p53RGs belonging to the p21-DREAM pathway. (**), p53RGs belonging to the MVA pathway.

ClQ_DEGs_#993 (wtp53)		
p53RGs (80) * p21-DREAM	**down-regulated** *anln *; arhgap11a *; arhgap11b *; aspm *; aurka *; bub1 *; bub1b *; casc5 *; c11orf82 *; c12orf48 *; ccdc18 *; ccna2 *; ccnb1 *; ccnb2 *; ccne2; cdc2; cdca2 *; cdca3 *; cdca8 *; cdc25c *; cdkn3 *; cenpe *; cit *; ckap2l *; dc; depdc1 *; depdc1b *; dlgap5 *; esco2 *; exo1 *, fam64a *; fancb *; fancd2 **; fanci **; gas2l3 *; gtse1 *; hist1h2bm *; hjurp *; hmmr *; kif2c *; kif4a *; kif11 *; kif14 *; kif15 *; kif18a *; kif20a *; kif20b *; kif23 *; kif24 *; mad2l1 *; melk *; mki67 *; ncapg *; ncapg2 *; ncaph *; ndc80 *; neil3 *; nuf2 *; nusap1 *; plk1 *; plk4 *; polq *; prc1 *; prr11 *; pttg1 *; racgap1 *; rrm2; rtkn2 *; sema3a; sgol1 *; sgol2 *; shcbp1 *; spag5 *; spc25 *; stil *; top2a *; tpx2 *; troap *; ttk *; ube2c *; xrcc2 **	**GO Terms:** cell cycle mitosis cytokinesis DDR
p53RGs (12) ** MVA	**up-regulated** *acat2 **; dhcr7 **; dhcr24 **; fasn **; fdft1 **; fdps **; lpin1 **; lss **; mvd **; nsdhl **; sc4mol **; tm7sf2 ***	**GO Terms:** lipid metabolism cholesterol- biosynthesis

**Table 2 cells-12-01290-t002:** Contribution of p53-regulated genes to the transcriptomic response induced by ClQ in GSCs expressing R273H-p53 mutant. Abbreviations as in Table 1. (*), p53RGs belonging to the p21-DREAM pathway. (**), p53RGs belonging to the MVA pathway.

ClQ_DEGs_G112 (R273H-p53)		
p53RGs (9) * p21-DREAM	**down-regulated:** *ccnb1 *; dlgap5 *; exo1 *, kif4a *; kif20a *; kif23 *; neil3 *; sema3a; top2a **	**GO Terms:** cell cycle mitosis cytokinesis DDR
p53RGs (7) ** MVA	**up-regulated:** *dhcr7 **; fasn **; fdps **; hmgcr **; hmgcs1 **; lss **; sc4mol ***	**GO Terms:** lipid metabolism cholesterol- biosynthesis

**Table 3 cells-12-01290-t003:** Contribution of p53-regulated genes to the transcriptomic response induced by ClQ in GSCs expressing R273H-p53 mutant. Abbreviations as in Table 1. (*), genes associated with GB promotion.

ClQ_DEGs_#1095 (p53 null)		
p53RGs (0) * GB promotion	**down-regulated:** *arhgap29 *; id1 *; id3 *; igfbp5 *; itga3 * tnfaip3 *; trdc*	**GO Terms:** receptor activity migration
p53RGs (0) * GB promotion	**up-regulated:** *acsl6 *; bhlhe41; cryab *; ddit4l *; fabp3 *; fn3k; gdf15 *; gpnmb *; lipg; lrrc39; nckap5 *; pcsk6 *; pfkfb2 *; pi15; pnliprp3; serinc5; st3gal5 **	**GO Terms:** proliferation metabolism

## Data Availability

Gene expression data and results of bioinformatic analyses are deposited in Gene Expression Omnibus database and can be found under the accession number GSE225191.

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
