# Peer review of "Concurrent Activation of Both Survival-Promoting and Death-Inducing Signaling by Chloroquine in Glioblastoma Stem Cells: Implications for Potential Risks and Benefits of Using Chloroquine as Radiosensitizer"

_cells, 2023, doi:10.3390/cells12091290_

Round 1

Reviewer 1 Report

In this submission, Andreas Mueller and colleagues explored the “atypical” mechanisms of the antimalarial agent chloroquine-driven radio sensitization of glioblastoma cells.  The topic is timely.  The effects of chloroquine, typically explained through its interactions with lysosomes and other low pH organelles, and inhibition of autophagy.  However, chloroquine also modifies plasma membrane stability and alters signaling pathways and transcription (e.g., Schrezenmeier & Dörner, 2020).  The relevant experiments have been conducted in GBM cell models and animal models.  Genetically diverse cell lines, differing in p53 status, have been employed, fortifying the clinical relevance of the presented work.

This study is generally well-planned and well-described.  However, a number of conclusions are not supported by quantitative information.  Some suggestions for improvement are enumerated below:

[1]  The types of statistical analyses (including ad hoc correction for multiple comparisons) and the unified approach for data replication and power analysis are not specified in the Methods section. This has to be rectified.   The exact size of experimental groups has to be provided in all figures.

[2]  Representative western blots in Figure 4 are insufficient for drawing scientific conclusions.  The relevant data have to be reproduced and presented in a quantitative manner.

[3] The above-mentioned concern is applicable to Figure 5.

[4] The above-mentioned concern is applicable to Figure 6.

[5] The above-mentioned concern is applicable to Figure 7.

[6] The p53-dependence of effects in Figure 8 needs to be put in the context of reproducibility across independently prepared and treated cell lines.  

[7] Date presented in Figure 9 are not replicated and therefore cannot be evaluated in the context of their thoroughness and reproducibility.

[8] The above-mentioned concern is applicable to Figure 10.

[9] The above-mentioned concern is applicable to Figure 11.

Author Response

We thank the Reviewer for his/her constructive feedback and suggestions, which we found very helpful for improving our manuscript. We have addressed all the points raised by the reviewer and hope to have met the expected requirements.  

1] The types of statistical analyses (including ad hoc correction for multiple comparisons) and the unified approach for data replication and power analysis are not specified in the Methods section. This has to be rectified. The exact size of experimental groups has to be provided in all figures 

This deficiency has now been corrected. Statistical analyses used in the study, size of experimental groups and number of experiments have now been specified in the Methods section and figure legends.

[2] Representative western blots in Figure 4 are insufficient for drawing scientific conclusions. The relevant data have to be reproduced and presented in a quantitative manner.

[3] The above-mentioned concern is applicable to Figure 5.

[4] The above-mentioned concern is applicable to Figure 6.

[5] The above-mentioned concern is applicable to Figure 7.

[8] The above-mentioned concern is applicable to Figure 10.

[9] The above-mentioned concern is applicable to Figure 11.

Representative data shown in Figs. 4-7, 10 and 11 have been reproduced in numerous experiments using cell cultures differing for the number of passages, different batches of antibodies and performed by different investigators from our group. We have now realized that this has not been explicitly stated in the first version of our manuscript, which might have led to the erroneous impression that the data shown in representative blots have not been replicated. We now explicitly state this in figure legends and provide a compilation of quantitative data from different experiments, in addition to representative blots shown in Figs. 4-7, 10 and 11. 

[6] The p53-dependence of effects in Figure 8 needs to be put in the context of reproducibility across independently prepared and treated cell lines.

To ensure the reproducibility of transcription-associated effects of chloroquine gene expression analyses were performed in three independently prepared and treated biological replicates of each cell line. This information has been provided in the GEO submission of gene expression data but not the manuscript. We thank the reviewer for pointing out this insufficiency and amend the description of gene expression analyses accordingly in the Methods section (lines 186-189). In the frame of this revision, we also provide the original data deposited to GEO for the reviewer’s perusal.

[7] Date presented in Figure 9 are not replicated and therefore cannot be evaluated in the context of their thoroughness and reproducibility.

The reviewer refers to our data obtained with the APOSIG arrays. These experiments have been instrumental in providing the initial hint that chloroquine reduces the abundance of AKT, a key factor in cancer radioresistance. We disagree with the reviewer’s view that these data “cannot be evaluated in the context of reproducibility” and argue that principal findings made with APOSIG arrays, namely that chloroquine affects the abundance of AKT has been confirmed and reproduced by Western blot (Fig. 10), which is a strong evidence for the validity of our APOSIG data. We consciously decided to reproduce APOSIG data by Western blot rather than engage in repetitive (and quite costly) experiments with APOSIG arrays for the following reasons: While being extremely useful as a “wide shot” approach in exploratory studies APOSIG arrays have some limitations, which preclude their utility in confirmatory investigations. Such as: i) antibodies spotted on APOSIG arrays recognize only phosphorylated forms of apoptosis-related proteins including AKT; ii) there is an uncertainty about binding epitopes of antibodies spotted on APOSIG arrays, this information is not released by the producer; iii) unlike in Western blot, specificity of antibodies binding or identity of protein target cannot be confirmed on the same array, which are not suitable for the validatory re-probing using another antibody binding the protein of interest. Due to these limitations repeating the experiments with APOSIG arrays would be of limited use for the unequivocal confirmation of initial observations that chloroquine has an impact on AKT. Therefore, in order to confirm and replicate the initial finding from APOSIG arrays, we chose Western blot as a more informative approach. This also turned out to be a good decision because it enabled us not only to validate the findings form APOSIG arrays but also gain further information that could not be obtained via APOGIC arrays.

Reviewer 2 Report

Cells 2023 -Müller et al.

The authors investigated the relationship between chloroquine-mediated radiosensitization and different p53 status in patient-derived glioblastoma stem cells using in vitro, in vivo and in silico approaches. They concluded that p53 functional status should be considered when evaluating the efficacy of glioblastoma radiosensitization by chloroquine. Mutated p53 was considered a positive predictive factor in chloroquine-mediated radiosensitization. They also demonstrated that in the absence of radiation, the potential effectiveness of chloroquine as monotherapy for glioblastoma is questionable or may lead to unwanted responses. The importance of in vivo testing was emphasized because the impacts of tumor microenviroment should not be neglected. The study was carefully undertaken with update methodology, thus being relevant from the point of view of obtaining results whose relevance extends from basic science to clinical potential application. The manuscript was very well written and documented.

Minor:

Introduction – line 95 – Please write “ionizing radiation” in full the first time “IR” is cited. IR is also the acronym of infrared.

Figs. 2, 6, 7, 10, and 11 – Please add R273H-p53, p53 null, and wtp 53 to G112SP, #1095 and #993 legends, respectively, like you did in Fig. 1.

Fig. 3. “Broken lines correspond to sham-treated control groups”?

Fig. 9.  Enlarge the letters to make them easier to read.

Fig. 10. Make the letters and numbers at the X and Y axes more readable.

Fig. 12. Turn the letters and lines of the boxes from gray to black or another color to make them more readable.

lines 330 and 400 – Western

line 404 - status

References: Standardize the format of the citations. Journal names: abbreviated or in full? PNAS or Proc Natl Acad Sci USA? Why some article titles were edited in italics?

References 82 and 85 are the same!

Correct: line 600 – “144”; line 606 – “343”; line 629 – “212”; lines 649, 662 and 669 – article number or page numbers is missing; line 656 – “324”; line 665 – “224”; line 667 – “78”; line 670 – “315”; line 696 – “1077”; line 727 – “1323”; line 733 – “3426”; line 746 – “738”.

Supplementary Figures: Legends are required.

Author Response

Thank you very much for your feedback and most helpful notices of several insufficiencies, which have obviously escaped our attention. All the deficiencies  specified in your comments have now been fixed. 

Introduction – line 95 – Please write “ionizing radiation” in full the first time “IR” is cited. IR is also the acronym of infrared.>>>  done

Figs. 2, 6, 7, 10, and 11 – Please add R273H-p53, p53 null, and wtp 53 to G112SP, #1095 and #993 legends, respectively, like you did in Fig. 1 >>> done

Fig. 3. “Broken lines correspond to sham-treated control groups”? >>> This is a typing error in the Legend for Fig. 3. Sham-treated control groups correspond to solid line curves. Thank you for pointing this error, it has now been corrected 

Fig. 9.  Enlarge the letters to make them easier to read >>> done

Fig. 10. Make the letters and numbers at the X and Y axes more readable. >>>done

Fig. 12. Turn the letters and lines of the boxes from gray to black or another color to make them more readable. >>>done

lines 330 and 400 – Western >>> corrected

line 404 – status  >>> corrected

References: Standardize the format of the citations. Journal names: abbreviated or in full? PNAS or Proc Natl Acad Sci USA? Why some article titles were edited in italics? >>>Citations format has been edited and standardized 

References 82 and 85 are the same! >>> thank you for pointing this inaccuracy, duplicated reference 85 has now been removed

Correct: line 600 – “144”; line 606 – “343”; line 629 – “212”; lines 649, 662 and 669 – article number or page numbers is missing; line 656 – “324”; line 665 – “224”; line 667 – “78”; line 670 – “315”; line 696 – “1077”; line 727 – “1323”; line 733 – “3426”; line 746 – “738”. >>> all corrected

Supplementary Figures: Legends are required.>>> According with the Cells format requirements legends to supplementary figures have been placed in the “Supplementary materials” section

Round 2

Reviewer 1 Report

I am satisfied with the Authors' responses.  The revised manuscript has been significantly improved and meets all criteria for publication.